# Enteric Methane Emission, Rumen Fermentation and Microbial Profiles of Meat-Master Lambs Supplemented with Barley Fodder Sprouts

Thamsanqa Doctor Empire Mpanza [1,*], Thabo Creswell Dhlamini [1,2], Rian Ewald Pierneef [3] and Khanyisile R. Mbatha [4]

[1] Agricultural Research Council-Animal Production: Animal Nutrition Section, Private Bag X2, Irene 0062, South Africa
[2] Department of Agriculture and Animal Health, Science Campus, University of South Africa, Private Bag X6, Florida 1710, South Africa
[3] Agricultural Research Council-Biotechnology Platform, Private Bag X05, Ondersterpoort 0110, South Africa
[4] College of Graduate Studies, School of Interdisciplinary Research and Graduate Studies, Muckleneuk Campus, University of South Africa, Pretoria 0003, South Africa
* Correspondence: mpanzat@arc.agric.za; Tel.: +27-12-672-9215

**Abstract:** This study evaluated the effects of barley sprout on the ruminal fermentation characteristics, enteric methane emission and microbiome profiles of meat-master lambs. Twelve uncastrated lambs aged 3 months were used. They were randomly assigned to three dietary treatments: *Eragrostis curvula* hay as a control diet (T1), grass hay plus 25% barley sprouts (T2) and grass hay plus 50% barley sprouts (T3). Animals were fed the diet for 61 days, including 10 days of adaptation. Four animals per treatment were used to collect methane and rumen fluid. Methane emission was recorded for nine consecutive days, from day 52 to 60, using a hand-held laser detector. Rumen fluid was collected on day 61 using an esophageal stomach tube for volatile fatty acid and DNA sequencing. The sprout supplementation had significant ($p < 0.05$) effects on methane emission and ruminal fermentation. Significant effects on rumen fermentation were observed with regards to ammonia–nitrogen ($NH_3$-N), acetic acid and a tendency ($p < 0.0536$) to increase propionic acid. Barley sprouts reduced methane gas emission, ammonia–nitrogen and the enhanced body weight of the animals. The bacteria *Bacteroidota* and *Firmicutes* were predominant among the identified phyla. In addition, there was a shift in the relative abundance of phylum among the treatments. The principal coordinate analysis showed a clear difference in microbiome among animals in T1 and those in T2 and T3. The sprout supplementation improves feed utilization efficiency by the animals. In conclusion, barley sprouts may be strategically used as a climate-smart feed resource for ruminants.

**Keywords:** hydroponic sprout; microbial community; methane mitigation; fermentation characteristics

## 1. Introduction

Ruminants are classified as animals with four stomach compartments. The rumen is the largest (ca. 10 L for sheep or goats) of the four compartments in the ruminant digestive system. The rumen is a dynamic, diverse and complex compartment, due to the presence of the consortia of microbes (bacteria $10^{10}$–$10^{11}$, archaea $10^8$–$10^9$, protozoa $10^5$–$10^6$ and fungi $10^3$–$10^4$ organisms/mL). These microbes play a crucial role in the rumen during the fermentation process [1–3]. Microbes such as bacteria, fungi and protozoa break down complex compounds (such as carbohydrates) into volatile fatty acids (VFAs), predominantly acetate, propionate and butyrate [4]. The produced VFAs in the rumen are the major source of energy for the animal [2,5]. Due to ruminal microbes, ruminant animals can convert fibrous plant material into an energy source [6]. Ruminant animals generate about 80% of the required energy for growth and production performance from the rumen

fermentation of fibers [7,8]. During the enteric fermentation, ruminants turn inedible coarse plant material into milk and meat [9]. Hence, ruminants are the only animals that are efficiently able to utilize cellulolytic and hemicellulolytic feedstuffs [1,10,11].

Because of rumen fermentation, ruminant animals are blamed for contributing to the production of anthropogenic greenhouse gas, enteric methane gas being the major concern due to its effect on the environment [12–15]. This is because microbes (e.g., archaea bacteria) utilize carbon dioxide ($CO_2$) and hydrogen gas ($H_2$) produced during fermentation as the main substrates for enteric methane production [4]. Enteric methane gas emitted by ruminants is 25 times more potent than $CO_2$ for the environment [16]. Moreover, enteric methane gas emitted by ruminants represents up to 12% of gross energy loss from the animal [17]. Therefore, there is a need to modulate rumen fermentation to reduce enteric methane gas emission without affecting feed utilization efficiency. Even though rumen microbiota are relatively stable, drastic changes have been reported due to feeding strategies and diets [18–20].

In this study, hydroponically produced fodder sprouts from barley (*Hordeum vulgare* L.) seeds were strategically fed to meat-master lambs. Sprouts are produced by transforming seed grains into a high quality and very lush fodder over a period of 7 to 10 days. The strategy does not use soil as the growing media, instead using a water or nutrient solution [21], hence the name *hydro*—(water/solution)—*ponic* (working). Therefore, this practice is regarded as an ideal solution in places where land is the major limiting factor for forage production [22]. Furthermore, a hydroponic fodder system is an ideal technology for arid and semiarid areas. Hydroponic farming has the potential of producing about 29 to 38.03 kg/$m^2$ of fresh fodder sprouts in a cycle of 8 days [23]. However, it is worth noting that hydroponic fodder sprouts should not be fed as the sole feed for ruminants because of a very low dry-matter content which is usually less than 20%.

Hydroponic fodder sprout contains bioactive catalysts (enzymes) which may help in feed digestion [24]. Moreover, the liquid from sprout is a potential source of nutrients for ruminal microbes [24], since sprouts contain above 80% moisture content. Feeding hydroponic fodder sprout to ruminants was reported to improve nutrient digestibility and increase ruminal enzyme activities [25]. The objective of this study was to determine enteric methane gas emission and the rumen fermentation profile, and characterize the rumen microbes of meat-master lambs as affected by barley fodder sprouts supplementation. It was hypothesized that supplementing meat-master lambs with barley fodder sprouts would affect enteric methane gas emission, rumen fermentation and microbiota.

## 2. Materials and Methods

### 2.1. Study Area and Ethics Approval

The study was conducted at Agricultural Research Council-Animal production (ARC-AP), geographically located at 25°53′53″ S; 28°11′25″ E, situated at 1480 m above sea level. This area has two distinct seasons: the dry season that extends from March to September and the short-wet season extend from October to February. The wet season is warm, while the dry season is cold (with some sporadic frost) and sunny between May and July.

The study protocol in using animals during the experiment was approved by ARC-AP, Animal Ethics Committee (Ref APAEC 2019/15) and the University of South Africa (UNISA), College of Agriculture and Environmental Sciences–Animal Research Ethics Committee (Ref 2021/CAES_AREC/064).

### 2.2. Hydroponic Barley Fodder Sprout Production

Barley (*Hordeum vulgare* L.) seeds used for hydroponic fodder sprouts production were purchased from Barenbrug SA Seeds (Pty) Ltd. Cape Town, South Africa, with a germination rate of roughly 80–89%. A portion of the storage house at ARC-AP, Nutrition Section, measuring 20 × 6.0 × 3.5 m (L × W × H), was used for hydroponic fodder sprout production. The portion of the house was cleared and nine metal stands each measuring 89 × 46 × 215 cm (L × W × H), with five shelves each, were installed. Each of the

metal stands could carry ten hydroponic plastic trays each measuring 41 × 28 × 5 cm (L × W × H), respectively. The metal stands were joined together to form a single row for stableness. Fodder sprouts were produced at room temperature with no other light other than natural sunlight. Hydroponic trays were washed with soap water and rinsed with tap water. Before planting, the trays were disinfected by soaking in a 1% sodium hypochlorite (NaClO) solution. The barley seeds (1 kg dry weight) were weighed into a separate container and soaked for 30 min in 10% NaClO solution [26]. After that, the seeds were rinsed with tap water three times and soaked again overnight in tap water. On the following day, wet barley seeds (about 2.1 kg wet mass) were spread on the hydroponic perforated plastic tray with a thickness of about 2.5 to 3 cm. Trays were irrigated manually with tap water using a 12 L Knapsack spray three times daily, at 09:00, 13:00 and 16:00. At harvest, day seven, to be fed on day eight, sprouts per tray weighed between 5.5 to 6 kg fresh biomass. A day (day seven) before lambs' feeding, sprouts were removed from the tray and allowed to drip off access water, and then hand shredded into small pieces for easy consumption by lambs. After shredding, three representative samples (300 g each) of sprouts were weighed and oven dried to determine the dry-matter content of sprout and kept for chemical composition analysis.

### 2.3. Study Animals and Diets

A total of 12 healthy and uncastrated male meat-master lambs about 3 months of age with the initial body weight of 23.3 ± 2.3 kg, were bought from a local supplier. Upon arrival at ARC-AP, animals were treated for internal and external parasites. Animals were weighed and randomly assigned to three dietary treatments, resulting in four animals per treatment. The dietary treatments were *Eragrostis curvula* grass (hereafter referred to as grass hay) as a control diet (T1), grass hay plus 25% hydroponic fodder sprouts on a dry-matter basis (T2) and grass hay plus 50% hydroponic fodder sprouts on a dry-matter basis (T3). A total of 25% and 50% of the hydroponic barley fodder sprouts (hereafter referred to as sprouts) were calculated from the daily intake of grass hay per animal and were offered in addition to grass intake. However, because of the high moisture (>80%) content in sprouts, grass hay and sprouts were offered separately. All animals received a daily concentrate supplement of 300 g and water was freely available throughout the study period. The concentrate was made of hominy chops (50%), wheat bran (36%), soybean meal (12%), feed lime (1.5%), salt (0.5%) and premix (1 pack). Animals were offered 3% of their body weight, of which 1% was from the concentrate while the remaining 2% was from grass hay for T1 diet and grass hay and sprout for T2 and T3 diets. The concentrate was offered once a day in the morning, at about 08:00, and was completely consumed by all animals. The nutritive value of grass hay, concentrate and barley sprout is indicated in Table S1 (Supporting Information). Animals finished the concentrate, thereafter the dietary treatment was offered per animal in a group. Animals were fed twice a day, at 08:00 and 16:00. Animals were maintained in their respective dietary treatment for 61 days, including 10 days for adaptation before data collection.

### 2.4. Data Collection

Feed intake and animal final body weight were recorded. Methane gas emission data were collected in all four animals per treatment, for nine consecutive days (i.e., from day 52 to 60). Methane data were recorded using the hand-held laser methane detector (LMD) machine (Crowcon Detection Instruments Ltd., Tokyo, Japan). The LMD equipment records methane concentration using a red laser beam pointed in the nostril of an animal, as described by Chagunda et al. [27]. Briefly, the methane gas was recorded by pointing a red laser beam at the nostril of lambs to estimate methane gas concentration at a distance of 1 m away from the animal. The recorded methane data in part per million (ppm) were

converted into g/day using the formula of Chagunda et al. [27], and it was calculated into g/kg BW and g/kg DMI.

$$M_{DG} = 0.000567 \times MTV \times TVr \tag{1}$$

where $M_{DG}$ is the daily methane in grams after including the conversion factor, MTV is the methane (ppm) recorded from the animal's nostril using hand-held LMD and TVr is the tidal volume of air when the animal was standing.

Rumen fluid (both solid and liquid fractions) was collected before morning feeding on day 61. The rumen fluid was collected using an esophageal stomach tube following the procedure of Shen et al. [28]. Briefly, the first 50 mL of rumen fluid was discarded due to possible salivary contamination. The second 50 mL was collected into a 100 mL container per animal. After collection, rumen fluid was divided into two halves; one half was used for rumen fermentation and the other half was used for DNA extraction. The samples for DNA extract were frozen and stored in a $-80\ °C$ freezer immediately, until further analysis.

### 2.5. Rumen Fermentation

Rumen fermentation was measured by the determination of ammonia–nitrogen and VFAs, as described by Wang et al. [29]. The rumen fluid was strained through three layers of cheesecloth. Then 4 mL of rumen fluid was acidified with 0.2 M HCl in a 1:1 acid: rumen fluid ratio and used to determine ammonia–nitrogen. In addition, another 5 mL of rumen fluid was acidified with 25% metaphosphoric acid in a 1:4 acid: rumen fluid ratio for the determination of VFAs.

### 2.6. Deoxyribonucleic Acid Extraction and Amplicon Sequencing

Microbial DNA was extracted using the Macherey-Nagel™ NucleoSpin™ DNA Stool kit, and 16S ribosomal RNA (16S rRNA) amplification and sequencing were performed according to the Illumina 16S protocol (16S Metagenomic Sequencing Library Preparation Guide). Briefly, the variable V3 and V4 regions of the 16S rRNA gene were amplified primers from Klindworth et al. [30] from the samples, followed by library amplification and sequencing on the Illumina MiSeq instrument using V3 chemistry. The primer sequence was as follows: 16S forward primer = 5′ TCGTCGGCAGCGTCAGATGTGTATAAGA-GACAGCCTACGGGNGGCWGCAG and 16S Reverse primer = 5′ GTCTCGTGGGCTCG-GAGATGTGTATAAGAGACAGGACTACHVGGGTATCTAATCC. The PCR program was as follows: 95 °C for 3 min, 25 cycles of; 95 °C for 30 s, 55 °C for 30 s, 72 °C for 30 s and a final extension at 72 °C for 5 min, held at 4 °C. Generated data were evaluated for quality and used for downstream bioinformatic pipelines. Low-quality sequencing reads were filtered and trimmed to a consistent length with a maximum of 2 expected errors per -read enforced [31]. This is done on paired reads jointly, after which amplicon sequence variants are inferred and downstream analysis is done using the DADA2 method [32]. This method combines identical sequencing reads into "unique sequences" with a corresponding abundance value followed by the identification of sequencing errors. Thereafter the forward and reverse reads are merged, and paired sequences that do not perfectly overlap are discarded.

The resulting sequence table was inspected for chimeras which were removed. Taxonomy was assigned to the final, filtered sequence table using the SILVA ribosomal RNA gene reference database [33]. The R package, phyloseq [34], was used to further analyze and graphically display the sequencing data which was clustered into amplicon sequence variants (ASVs) with the protocol described above. The ASV table was agglomerated onto operational taxonomic units (OTUs) according to taxonomic classification and inspected at "phylum" level to remove any unclassified OTUs.

The OTU table was normalized using the 'normalize function' and the 'median ratio" method implemented in MetalonDA R package [35] which uses the DESeq2 "estimate size factors" function [36]. For this analysis, we added a pseudo count of 1 to the initial OTU table, running the normalization prior to rounding off the normalized table to the largest integer not exceeding the normalized value. Floor rounding was applied to negate the

effect of the pseudo-count addition. The sequencing dataset was deposited at the National Center of Biotechnology Information (NCBI) Sequence Read Archive (SRA) database under BioProject ID: PRJNA865290.

### 2.7. Data Analysis

Data on feed intake, weight gain, enteric methane emission and fermentation profile were subjected to analysis of variance (ANOVA) using SAS software [37]

All bioinformatic analysis was done using a RStudio environment [38] with R core ream, version 4.0.2, Veinna, Austria [39]. Alpha diversity indices (i.e., Observed, Chaol, ACE, Shannon, Simpson, InvSimpson and Fisher) were tested for equal variances using the Bartlett test of homogeneity of variances available from the stats version 4.1.2 package implemented in R. Differences in alpha diversity values between the treatments for all the indices was calculated and visualized using the "ggbetweenstats" function available from the ggstatsplot v. 0.9.3 package. Alpha diversity value differences between treatments were tested using a one-way analysis of means and pairwise t-test after homogeneity of variances was confirmed with Bartlett's test. Benjamini–Hochberg (BH), which is the same as false discovery rate (FDR) in R, *p*-value adjustment was used for multiple comparisons. Principal coordinate analysis (PCoA) was performed using the ordinate function from phylose version 1.38.0 with "gower" distance metric specified [40]. To determine whether the variation exists between the treatments compared to within the treatments, anosim analysis was conducted by R package vegan version 2.6.2 [40] and "gower" distance metric was used to measure dissimilarity between samples.

For all other statistical tests, differences between the means were considered significant at $p \leq 0.05$ and a trend was declared at $0.05 < p \leq 0.10$. In case of significant difference between the means, LSMEANS procedure of SAS [37] software was used to separate the means. Pearson's correlation analysis was conducted to assess the relationship between the abundance of microbiota and volatile fatty acids, methane emissions and body weight using PROC CORR procedure of SAS program.

## 3. Results

### 3.1. Methane Emission as Influenced by Barley Sprout Supplementation

Data on performance in terms of daily feed intake, final body weight and enteric methane emission of meat-master lambs are shown in Table 1. Barley sprouts supplementation to meat-master lambs significantly ($p = 0.0186$) increased the feed intake and body weight of the meat-master lambs, while significantly reducing enteric methane emission. Animals in T2 and T3 (supplemented with barley sprout) ate 41% and 67% more feed, respectively, as compared to the T1 group. Similarly, animals in T2 and T3 were 11% and 17% heavier, respectively, than the T1 group. Supplementing meat-master lambs with barley reduced the methane yield per body weight ($p < 0.0264$) and per feed intake ($p < 0.001$) significantly. Methane gas emission on grams per feed intake was reduced by 34% and 52%, respectively, for animals in T2 and T3, as compared to the T1 group.

**Table 1.** Effects of supplementing barley sprouts on methane emission of meat-master lambs.

| Parameters | Treatments | | | SEM [1] | *p*-Value |
| --- | --- | --- | --- | --- | --- |
| | T1 | T2 | T3 | | |
| Intake (kg) | 1.0 [c] | 1.4 [b] | 1.6 [a] | 0.05 | <0.0001 |
| Initial body weight (kg) | 23.2 | 23.3 | 23.5 | 1.16 | 0.9871 |
| Final body weight (kg) | 28.6 [b] | 31.9 [ab] | 33.5 [a] | 1.12 | 0.0186 |
| Methane (ppm) [2] | 23.4 | 22.0 | 19.1 | 1.11 | 0.0608 |
| Methane (g/day) | 51.1 | 48.2 | 41.9 | 2.44 | 0.0608 |
| Methane (g/kg BW) [3] | 1.8 [a] | 1.5 [ab] | 1.3 [b] | 0.11 | 0.0264 |
| Methane (g/DMI) [4] | 53.0 [a] | 35.1 [b] | 25.5 [c] | 2.50 | <0.0001 |

[a,b,c] letters in superscript within a row means different significant at $p \leq 0.05$, trend was declared at $0.05 < p \leq 0.10$: [1] = standard error of the mean; [2] = part per million, [3] = body weight, [4] = dry-matter intake.

Figure 1 shows daily enteric methane emission of meat-master lambs as influenced by barley sprouts supplementation. The trend or the pattern of enteric methane emission recorded in animals consuming different dietary treatments was the same.

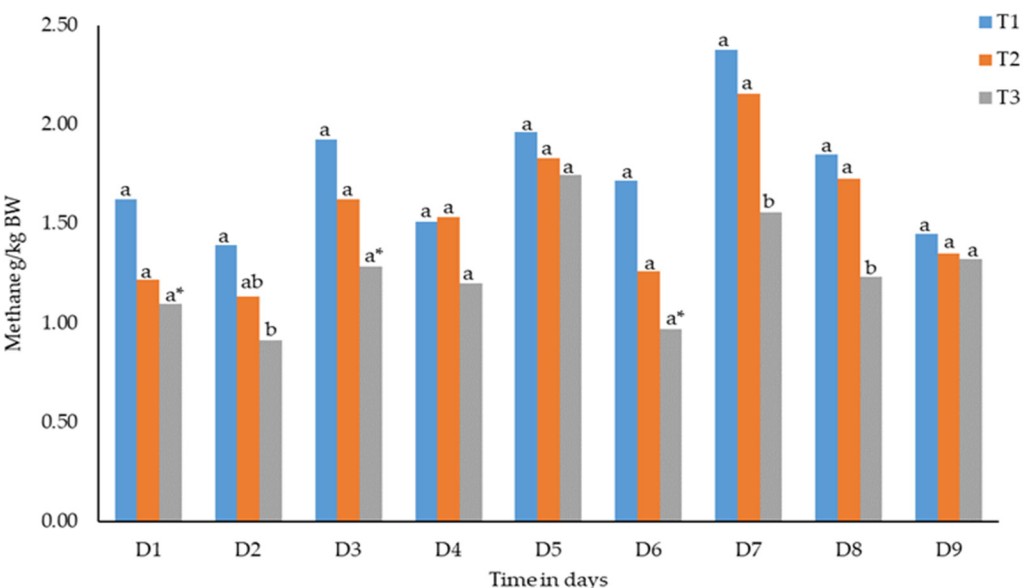

**Figure 1.** Effects of barley sprouts supplementation on daily methane emission potential of meat-master lambs. "a, b" letters within a day means different significantly at $p \leq 0.05$ and "a*" denotes the trend at $0.05 < p \leq 0.10$.

### 3.2. Rumen Fermentation Profile

Table 2 shows the rumen fermentation profile of meat-master lambs influenced by barley sprouts supplementation. Barley sprouts supplementation significantly ($p < 0.0001$) decreased the levels of ammonia–nitrogen ($NH_3$-N). Meat-master lambs that were in T2 and T3 had 42.8% and 48.5% less $NH_3$-N, respectively as compared to the T1 group. There was no significant ($p > 0.05$) variation observed in total volatile fatty acid (FVA), butyric, iso-butyric, valeric and iso-valeric acids concentration in the rumen fluid. Iso-butyric acid was not detected in rumen fluid of animals that were in T3. However, there was a significant ($p < 0.05$) decrease in acetic acid while there was a tendency ($p = 0.0536$) in increasing propionic acid. Animals that were in T2 and T3 produced 8.2% and 7.5% less acetic acids as compared to animals that were in T1. Moreover, there was a significant ($p = 0.0286$) decrease in acetic and propionic ratio observed in animals that were supplemented with barley sprouts. Animals that were fed in T2 and T3 had 24% and 26% lower acetic/propionic ratios, respectively as compared to animals in T1.

### 3.3. Rumen Microbial Composition of Meat-Master Lambs as Influenced by Barley Sprouts Supplementation

A total of 272 OTUs were obtained from all the samples. The rarefaction curves suggest that sampling of the rumen environments has a sufficient sequencing depth across the treatments (data not shown). A total of 24 phyla, 40 classes, 68 orders, 96 families and 166 genera were identified in rumen fluid collected from all three groups of animals that were fed different dietary treatments. There was no statistically significant ($p > 0.05$) differences detected in alpha diversity for Observed, Chao1, ACE, Shannon, Simpson, InvSimpson and Fisher (data not shown). Figure 2 shows the relative abundance of bacteria at the phylum level above 1% at least in one group. However, the relative abundance of all bacterial phylum that was detected in rumen fluid of meat-master lambs as influenced by sprouts supplementation are presented in Table S2 (Supporting Information). The most predominant bacterial phyla that were identified in the rumen fluid of meat-master lambs were *Bacteroidota* and *Firmicutes* and they constituted an amount of 73% of

the total bacterial phyla identified. The relative abundance of *Bacteroidota* phylum was 37.3%, 14.9% and 29.1% in T1, T2 and T3, respectively, while that of *Firmicutes* phylum was 27.0%, 52.9% and 35.4% in T1, T2 and T3, respectively. There was an observed variation of these bacterial phyla abundancy among the treatments; however, the variation was not statistically significant ($p > 0.05$).

**Table 2.** Effects of supplementing barley sprouts on the ruminal fermentation profile of meat-master lambs.

| Parameters | Treatments | | | SEM [1] | *p*-Value |
|---|---|---|---|---|---|
| | T1 | T2 | T3 | | |
| NH$_3$-N (mg/dL) [2] | 19.4 [a] | 11.1 [b] | 9.8 [b] | 0.66 | <0.0001 |
| Total VFA (mmol/L) [3] | 68.9 | 68.3 | 60.1 | 5.90 | 0.5344 |
| Molar proportion of VFA | | | | | |
| Acetate (A) | 73.3 [a] | 67.3 [b] | 67.8 [b] | 1.23 | 0.0286 |
| Propionate (P) | 16.5 | 20.1 | 20.8 | 1.04 | 0.0536 |
| Butyrate | 7.1 | 8.8 | 9.0 | 0.71 | 0.2068 |
| Iso-butyrate | 1.2 | 1.5 | nd [4] | 0.09 | 0.2671 |
| Valerate | 0.8 | 1.0 | 0.8 | 0.06 | 0.1276 |
| Iso-valerate | 1.2 | 1.3 | 1.0 | 0.18 | 0.4426 |
| A:P ratio | 4.5 [a] | 3.4 [b] | 3.3 [b] | 0.28 | 0.0408 |

[a,b] letters in superscript within a row means different significance at $p \le 0.05$; trend was declared at $0.05 < p \le 0.10$.
[1] = standard error of the mean, [2] = Ammonia–nitrogen, [3] = volatile fatty acid, [4] = not detected.

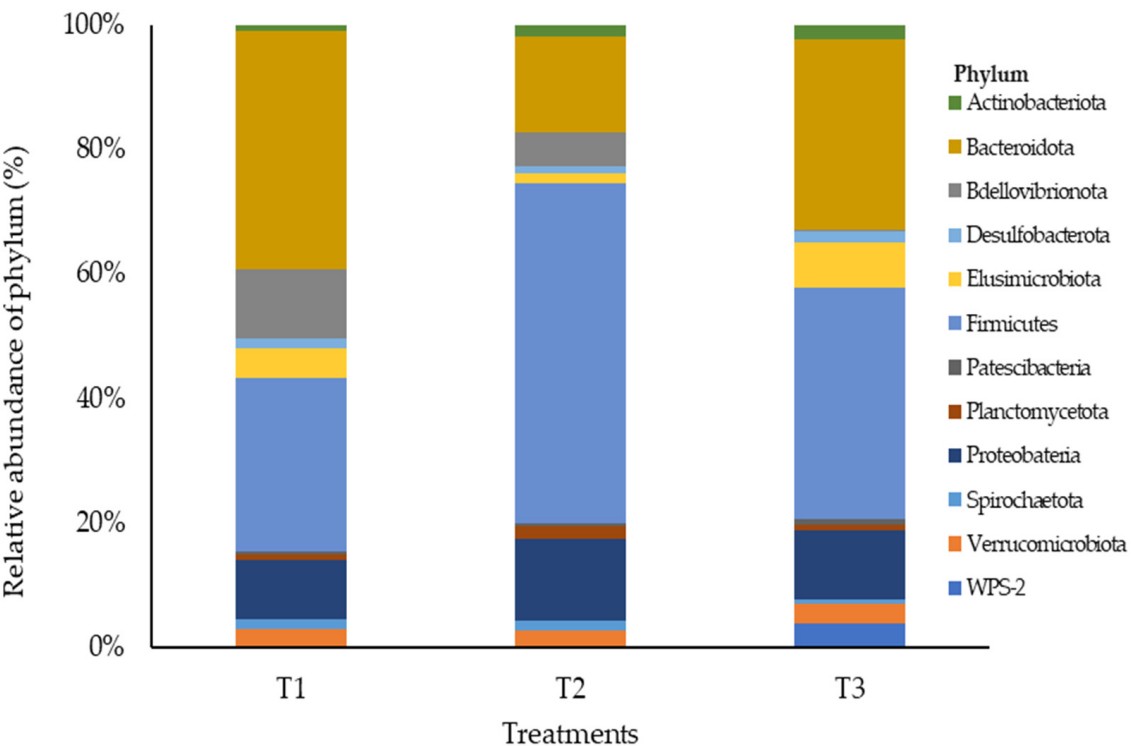

**Figure 2.** Bacterial relative abundance (>1% at least in one group) at phylum level as influenced by the sprout supplementation on meat-master lambs.

Table 3 shows the abundancy of bacteria at the family level. There were 25 families with a relative abundance of above 1% at least in one of the groups. The most predominant bacteria that were detected in animals that were on T1 were *p-2534-18B5 gut group* (26.54%), *Lachnospiraceae* (7.90%) and *Rikenellaceae* (7.83%). On animals that were fed T2, the dominating bacteria were *Erysipelatoclostridiaceae* (24.67%) and *Lachnospiraceae* (10.96%). In animals that were fed T3, the predominant abundant bacteria were *Prevotellaceae* (26.24%), *Lachnospiraceae* (12.36%) and *Selenomonadaceae* (10.53%). The Prevotellaceae that dominate

animals in T3 belonged to phylum *Bacteroidota*. There was a variation of these bacteria across the dietary treatments; however, the observed variation was not statistically significant ($p > 0.05$). A significant ($p < 0.05$) decrease in bacterial abundance as an influence of sprouts supplementation was observed in *Butyricicoccaceae* (1.0, 0.2 and 0.1%) and *Hungateiclostridiaceae* (4.2, 0.8 and 0.7) in T1, T2 and T3, respectively. A decreasing trend ($p = 0.0924$) was observed in *p-2534-18B5 gut group* bacteria, with an abundance level dropping from 26.5% to 0.2% as the sprouts supplementation increases.

**Table 3.** Effects of sprouts supplementation on bacterial abundance at family level on meat-master lambs (>1% at least in one group).

| Bacteria | Treatments | | | SEM [1] | *p*-Value |
|---|---|---|---|---|---|
| | **T1** | **T2** | **T3** | | |
| *Acholeplasmataceae* | 2.6 | 0.5 | 0.5 | 0.67 | 0.1097 |
| *Anaerovoracaceae* | 1.3 | 2.4 | 2.3 | 0.62 | 0.4303 |
| *Butyricicoccaceae* | 1.0 [a] | 0.2 [b] | 0.1 [b] | 0.20 | 0.0388 |
| *Atopobiaceae* | 0.3 | 0.9 | 1.8 | 0.47 | 0.1529 |
| *Chitinophagaceae* | 0.5 | 1.9 | 1.0 | 0.93 | 0.5826 |
| COB P4-1 termite group | 3.3 | 0.1 | 0.2 | 1.05 | 0.1246 |
| *Desulfovibrionaceae* | 1.0 | 0.9 | 1.9 | 0.70 | 0.5806 |
| *Erwiniaceae* | 1.5 | 0.2 | 0.2 | 0.81 | 0.4529 |
| *Erysipelatoclostridiaceae* | 1.5 | 24.7 | 1.5 | 13.09 | 0.4084 |
| *Erysipelotrichaceae* | 0.7 | 1.2 | 2.6 | 0.55 | 0.1279 |
| *Hungateiclostridiaceae* | 4.2 [a] | 0.8 [b] | 0.7 [b] | 0.64 | 0.013 |
| *Lachnospiraceae* | 7.9 | 11.0 | 12.4 | 2.44 | 0.4656 |
| *Moraxellaceae* | 1.2 | 1.3 | 0.5 | 0.68 | 0.7049 |
| *Oscillospiraceae* | 6.0 | 4.6 | 3.6 | 1.93 | 0.6761 |
| *Oxalobacteraceae* | 0.2 | 1.4 | 0.8 | 0.56 | 0.4259 |
| *p-2534-18B5 gut group* | 26.5 | 0.2 | 0.2 | 7.98 | 0.0924 |
| *Pasteurellaceae* | 1.7 | 1.3 | 1.9 | 0.89 | 0.8862 |
| *PeH15* | 2.7 | 5.5 | 0.2 | 3.16 | 0.5275 |
| *Pirellulaceae* | 1.0 | 1.9 | 0.6 | 0.46 | 0.1884 |
| *Prevotellaceae* | 1.5 | 5.9 | 26.2 | 12.10 | 0.3677 |
| *Rikenellaceae* | 7.8 | 0.8 | 0.8 | 3.96 | 0.4087 |
| *Ruminococcaceae* | 1.7 | 1.7 | 1.9 | 0.44 | 0.9587 |
| *Selenomonadaceae* | 0.9 | 4.6 | 10.5 | 3.90 | 0.2895 |
| *Spirochaetaceae* | 1.1 | 1.7 | 0.6 | 0.64 | 0.5574 |
| *Succinivibrionaceae* | 2.1 | 5.7 | 4.9 | 3.25 | 0.7317 |

[a,b] letters in superscript within a row means different significant at $p \leq 0.05$, trend was declared at $0.05 < p \leq 0.10$; [1] = standard error of the mean.

Figure 3 shows the differences in rumen microbial communities of meat-master lambs as influenced by the sprout supplementation. According to the principal coordinate analysis (PCoA), there is a distinct separation in the bacterial communities between the animals subjected to different dietary treatments. The PCoA showed that the rumen microbial differences accounted for a 29.1% variation in the animals in the T1 diet, distinguished from animals in the T2 and T3 diets by axis 1, while rumen microbial differences between the animals that were in T2 and T3 represent a 23.2% variation of axis 2. The ANOSIM analysis confirmed that there was a significant ($p < 0.015$) dissimilarity of rumen bacteria in animals on the T1 diet in relation to the other two groups (Figure 4).

Correlation analysis was conducted to determine the correlation between the relative abundancy of microbial bacteria, fermentation parameters, methane emission and body weight. Table S3 (Supplementary Information) shows the Pearson's correlation analysis. There was a significant ($p < 0.0436$) negative correlation between phylum *Acinabacteriota* and ammonia–nitrogen ($NH_3$-N). Phylum *Desulfobacterota* had a significant ($p < 0.0065$) negative correlation with body weight and a significant ($p < 0.0124$) positive correlation with acetic and propionic ratio. *Frimicute* was positively corrected with valeric significantly ($p < 0.0402$). *Patescibacteria* negatively correlated with iso-butyrate and iso-varate significantly with

*p* < 0.0034 and *p* < 0.0239, respectively. *Planctomycetota* had a significant (*p* < 0.0219) positive correlation with valeric acid.

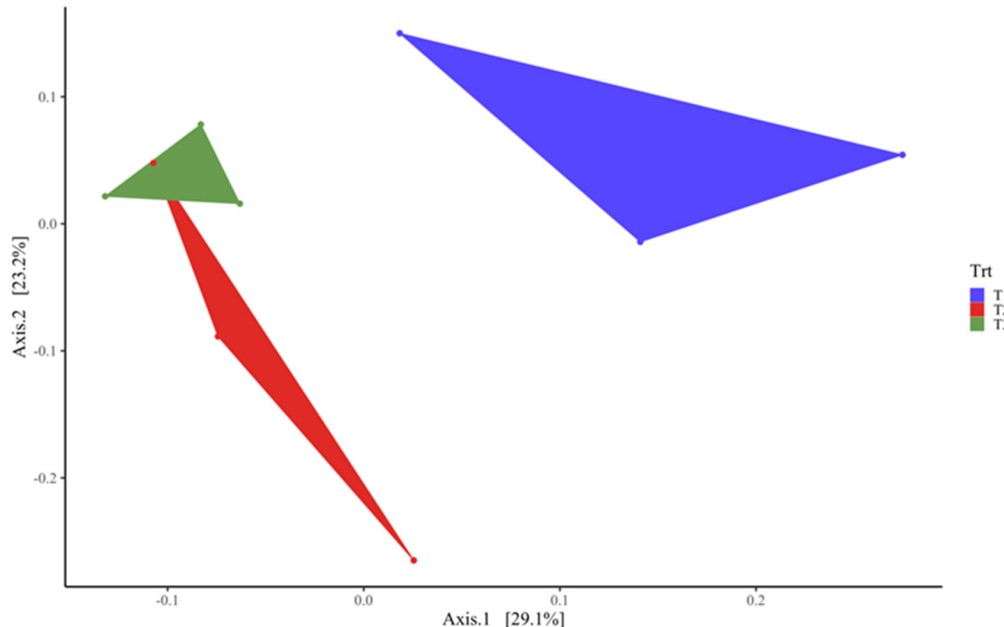

**Figure 3.** Principal coordinate analysis (PCoA) showing the differences in the bacterial community of meat-master lambs as influenced by sprouts supplementation.

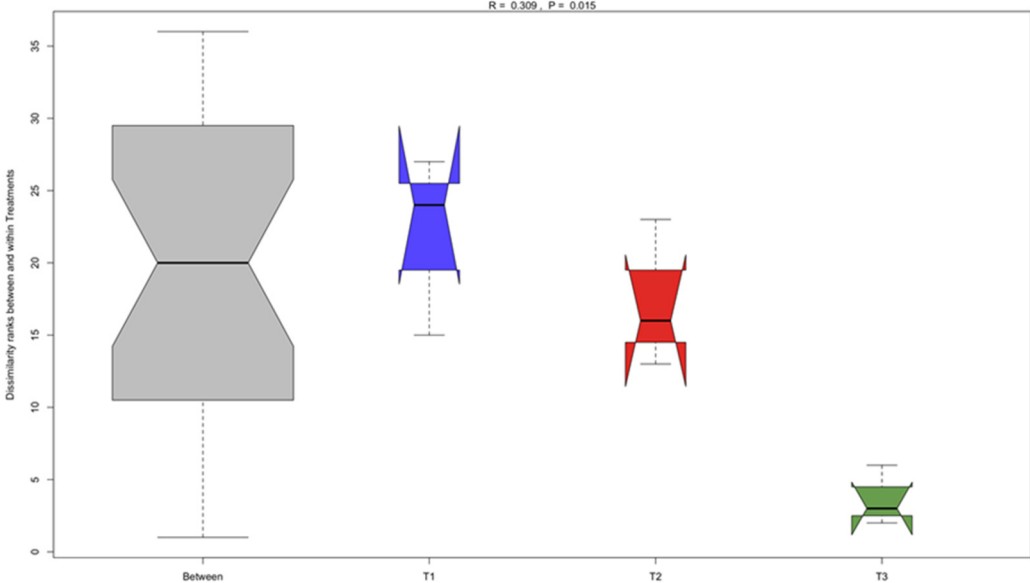

**Figure 4.** ANOSIM showing the dissimilarity between treatments.

## 4. Discussion

The rumen is the natural fermentation tank where fibrous materials and protein in the feed are degraded to produce volatile fatty acids and ammonia–nitrogen. Hence, rumen fermentation plays a critically important role in nutrient digestion and metabolism in ruminant animals [29]. On the other hand, diet type and diet quality play a huge role in the fermentation end production in the rumen [3,5,29]. For example, a forage-based diet has been associated with the high methane gas emission of ruminants [41]. This study was conducted in order to evaluate the effect of barley sprouts supplementation on the methane gas emission, ruminal fermentation and microbial dynamics of meat-master lambs.

### 4.1. Methane Emission and Ruminal Fermentation as Influenced by Barley Sprouts Supplementation

Feeding hydroponic barley sprouts to ruminants have been reported to improve nutrient digestibility and improve ruminal enzyme activities [25]. This is because sprouts contain enzymes and the liquid as the source of nutrients for ruminal microbes [24]. Therefore, supplementing barley sprouts to meat-master lambs could provide added and easily accessible nutrients for rumen microbes. This could be attributed to the reduction in methane gas emission observed in this study as sprouts supplementation increases. Methane gas emission depends on the feed quality and degradability of the fiber from the feed source. The longer it takes for the microbes to degrade fiber in the rumen, the more hydrogen ($H_2$) and carbon dioxide ($CO_2$) are produced. Kumar et al. [41] reported that a forage-based diet is associated with increased methane emissions when compared with a concentrate-based diet, due to an increased level of $H_2$ and $CO_2$ production. The accumulated $H_2$ and $CO_2$ in the rumen are used by methanogenic microbes to produce methane gas as a way of disposing $H_2$ from the rumen [4,42,43]. Despite the negative impact of methane gas on the environment, methane gas emitted by ruminants represent an energy loss for the animal and it has a negative impact on animal production. In this study, animals that produced less methane gas were 11% (T2) and 17% (T3) heavier than animals that produced more methane gas (T1). These results agree with Congio et al. [44], who reported in the meta-analysis review a reduction in methane emission while animal production (i.e., milk or daily weight gain) increases, simultaneously. In this study, animals that had higher feed intake produced less methane gas (Table 1). However, these results are not in agreement with Gaviria-Uribe et al. [45], who reported that zebu steers that had higher feed intake produced more methane gas. The discrepancy in the results reported between the two studies could be attributed to the feed that was consumed by animals, since feed influences rumen fermentation [3]. The reduction in methane-gas emission reported in the current study while feed intake increases can be associated with the efficient utilization of feed by animals due to sprouts supplementation. Reducing enteric methane emission from ruminants while enhancing feed conversion efficiency and nutrient utilization is the main goal for sustainable livestock production [46].

Supplementation of barley sprouts to meat-master lambs resulted in the reduction in ammonia–nitrogen ($NH_3$-N). However, the $NH_3$-N values that are recorded in this study are within the range of the 8.5 mg/dL minimum and 30 mg/dL maximum requirement for rumen microbial growth [47]. In the current study, the increase in barley sprouts supplementation reduces $NH_3$-N mean concentration significantly ($p < 0.0001$) from 19.4 mg/dL for T1 (control) to 11.1 and 9.8 mg/dL in animals in T2 and T3, respectively. According to Teklebrhan et al. [43], fermentation of protein in the rumen produces ammonia, therefore, this result indicates that barley sprouts supplementation reduced protein fermentation in the rumen. Panyawoot et al. [48] associated the decrease in $NH_3$-N on Thai Native-Anglo-Nubian goats fed fermented discarded durian peel with the reduction in protein fermentation. Teklebrhan et al. [43] associated the decrease in $NH_3$-N in goats fed corn gluten diet with the increased utilization of amino acids by the microbial community. The decrease in $NH_3$-N reported in this study can be associated with the efficient use of protein in the feed by meat-master lambs fed diet T2 and T3. However, barley sprouts cannot be supplemented by more than 50%, due to the possibility of reducing $NH_3$-N to below the minimum required level for microbial growth.

The VFAs are a source of energy for ruminant animals [2,5] as they provide roughly 80% digestible energy [7,8]. In the current study, barley sprout supplementation did not have a significant effect on VFA and butyric acid. The VFA values recorded in this study were below the range of 70 to 150 mmol/L values recommended by McDonald et al. [47]. On the other hand, sprout supplementation decreased acetic acid significantly, while it tended to increase propionic acid. Since propionic acid is the main source of glucose (energy) for ruminants [49], this tendency to increase propionic acid could have provided an increased energy availability to the animals in T2 and T3 diets. In addition, the ratio between

acetic and propionic acids was significantly reduced by barley sprouts supplementation. Ma et al. [50] reported that a lower ratio between acetic and propionic acids provides more energy for the host animal, which is beneficial for the growth of the animals. Therefore, this indicates that animals that were in T2 and T3 diets had more energy which was used for growth hence, they weighed 3.3 and 4.9 kg more than animals in the T1 diet. The reduction in acetic acid and the ratio between acetic and propionic acids could be associated with the reduction in methane gas emission from animals that were fed T2 and T3 diets. This is in agreement with the result of Morgavi et al. [51], who report a decrease in acetic to the propionic acid ratio in cattle, as influenced by *Manoscus* spp.

*4.2. Ruminal Microbial Composition as Influenced by Barley Sprouts Supplementation*

Ruminal microbiota plays a critical role in the maintenance, immunity and growth performance of ruminant animals [52]. Despite the stable nature of rumen microbes, dietary changes have been reported to be responsible for fluctuations in ruminal microbiota [19]. For example, a fibrous diet or concentrate to forage ration has been reported to have a great influence on ruminal microbiota changes [53,54]. In this study, the bacterial phyla *Bacteroidota* and *Firmicutes* were dominant in meat-master lambs. These results are in agreement with Bi et al. [55], Li et al. [3], Liu et al. [56], Linde et al. [57], Ahmad et al. [8] and Guo et al. [5], who reported the same phyla to be dominant in the rumen of a dairy cow, Tibetan sheep, Bonsmara cattle, *Bos grunniens*, and goat kids, respectively. The study conducted by Mani et al. [58] on meat-master sheep supplemented with direct-fed lactic acid bacteria showed also that *Bacteroidota* and *Firmicutes* were the most dominating phyla. Dominancy of *Bacteroidota* and *Firmicutes* in the rumen promotes forage fermentation [56]. Hence, these bacterial phyla are important in assessing the energy requirement of a ruminant animal [59]. The dominancy of *Bacteroidota* bacteria in the rumen has been associated with the high roughage or low concentrate diet [8,60]. According to Zhang et al. [61], *Bacteroidota* bacteria are responsible for the decomposition of starch and soluble carbohydrate, while *Firmicutes* is known to utilize xylan, cellulose and hemicellulose as energy sources [62].

Although *Bacteroidota* and *Firmicutes* were the dominant phyla in the rumen fluid of meat-master lambs, there was a shift in their relative abundance among the treatments. The stacked bar-plot (Figure 2) shows a clear shift of bacterial phylum abundance from *Bacteroidota* in animals that were in T1 to *Firmicutes* in animals that were in sprouts supplement (T2 and T3 animals). *Bacteroidota*-dominated animals were in the T1 group, while *Firmicutes*-dominated animals were supplemented with barley sprouts (T2 and T3). The changes in taxonomic composition at a phylum level influence the beta-diversity between treatments and as such results in clear clustering of the treatment groups. The principal coordinate analysis (Figure 3) showed discrete clustering of rumen bacteria of animals that were in the T1 diet from animals that were in T2 and T3 diets. The ANOSIM analysis (Figure 4) shows a significant ($p < 0.015$) dissimilarity of rumen bacteria between the treatments. ANOSIM analysis is used to determine whether a significant difference exits between treatments, compared to within treatments [63]. The change in bacterial dominancy could be influenced by sprouts supplementation and it is hypothesized that the enzymes and/or nutrients from sprouts are responsible for this change. Difference in the dietary treatment offered to animals has been reported to affect the ratio between *Firmicutes* and *Bacteroidota* [29]. Similarly, in this study, the supplementation of sprout to meat-master lambs has resulted in the shift of ruminal phylum abundance from *Bacteroidota* to *Firmicutes*. This shift has led to an increase in the *Firmicutes* to *Bacteroidota* ratio to 8.4 and 2.3 of animals in T2 and T3 diets, respectively, as opposed to 0.8 of animals in the T1 diet (Table S2 supporting information). The increase in the relative abundance of *Firmicutes* phylum increases the *Firmicutes* to *Bacteroidota* ratio, which is associated with efficient use of feed in animals [8,64]. Therefore, the abundance of *Firmicutes* in animals that were in T2 and T3 diets suggests the efficient use of feed by these animals. *Firmicutes* is reported to have an important function in the process of energy absorption [65]. Therefore, this study indicates that animals that were in T2 and T3 diets had a higher energy utilization,

that provides more energy for the host to improve weight gain. In addition, these animals produced significantly less methane per kg of feed intake in relation to animals in the T1 diet (Table 1), which indicates that they efficiently utilized the feed.

The shift in relative abundance of bacterial phyla observed in this study could also be associated with the decrease in $NH_3$-N observed in Table 2. This is because *Bacteroidota* abundance in the rumen is reported to have a positive correlation with the synthesis of $NH_3$-N [54]. The phylum *Bacteroidota* is able to efficiently break down protein and hence assist in the proteolysis process [66]. At the family level, the bacteria that were found to be dominant in animals in different dietary treatments were also related to the dominant phyla. For example, the *p-2534-18B5 gut group* bacteria that was dominant in the T1 group belong to the phylum *Bacteroidota*. The decreasing trend observed could be attributed to the shift of relative bacterial abundance from *Bacteroidota* to *Firmicutes* (Figure 2). Similarly, the animals in T2 were dominated by *Erysipelatoclostridiaceae* which belongs to the phylum *Firmicutes*. Surprisingly, animals in T3 were dominated by *Prevotellaceae* bacteria which belong to the phylum *Bacteroidota*. The two bacteria at the family level that were observed to decrease significantly as sprouts supplement increased, belonged to the *Firmicutes* phylum.

In this study, the Pearson's correlation analysis shows that there was a correlation between *Firmicutes* and valerate while there was no correlation with other VFAs. The lack of significant correlation with others VFAs does not suggest that those VFAs are not important. *Ferimicutes* phylum is reported to be involved in metabolism in the rumen [65], therefore, the expectation was that *Firmicute* will have a positive correlation with VFAs.

## 5. Conclusions and Recommendations

This study showed that barley sprouts modulated the rumen of meat-master lambs by affecting the bacterial profile and rumen fermentation. This has resulted in the change of bacterial relative abundancy and reduction in methane yield per kg of feed intake in animals that were fed barley sprouts. Therefore, this study showed that barley sprouts have the potential of improving animal-feed use efficiency and performance while reducing methane yield per feed intake. However, further study is required in order to characterize enzymes and nutrients in sprouts and their role in rumen modulation. This will provide a better understanding of how sprouts can be strategically used in ruminant feeding as climate-smart fodder.

**Supplementary Materials:** The following supporting information can be downloaded at: https://www.mdpi.com/article/10.3390/fermentation8090434/s1: Table S1: chemical composition of grass hay, concentrate and barley sprout use during the experiment; Table S2: relative abundance of rumen bacteria at phylum level of meat-master lambs and the ratio between *Bacteroidota* and *Firmicute phyla*; Table S3: Pearson correlation analysis between relative abundance of bacterial phyla and VFAs, methane and body weight.

**Author Contributions:** Conceptualization, T.D.E.M.; methodology, T.D.E.M., T.C.D. and K.R.M.; funding acquisition, T.D.E.M.; conducting the study, T.C.D. and T.D.E.M. validation, T.D.E.M., T.C.D. and K.R.M.; statistical analysis, R.E.P. and T.D.E.M.; resources, T.D.E.M.; data curation, T.D.E.M., K.R.M., R.E.P. and T.C.D.; writing—original draft preparation, T.D.E.M.; writing—review and editing, K.R.M., R.E.P. and T.C.D.; project administration, T.D.E.M.; All authors have read and agreed to the published version of the manuscript.

**Funding:** Gauteng Department of Agriculture and Rural Development (GDARD) funded this research; the grant number is P02000180.

**Institutional Review Board Statement:** This study was conducted under the approval of Animal Ethics Committee of Agricultural Research Council (ARC) and University of South Africa (UNISA), with the approval no APAEC 2019/15 and 2021/CAEC_AREC/064, respectively.

**Informed Consent Statement:** Not applicable.

**Data Availability Statement:** The datasets presented in this study were deposited in National Center of Biotechnology Information (NCBI) Sequence Read Archive (SRA) database under BioProject ID: PRJNA865290.

**Acknowledgments:** The authors would like to thank the ARC-Microbiology section for their assistance in collecting rumen fluid, the Beef Breeding section for borrowing the methane detector and the Meat Science at ARC who helped in storing the rumen fluid samples at −80 °C before laboratory analysis.

**Conflicts of Interest:** The authors declare no conflict of interest.

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
