# Peer review of "Enteric Methane Emission, Rumen Fermentation and Microbial Profiles of Meat-Master Lambs Supplemented with Barley Fodder Sprouts"

_fermentation, doi:10.3390/fermentation8090434_

Round 1

Reviewer 1 Report

This study investigated the effects of barley sprout on ruminal fermentation characteristics, enteric methane emission and microbiome profiles of meat-master lambs. The result showed barley sprout feeding decreased the methane emission, increased the rumen fermentation and growth performance, and changed the microbial community, which meant that barley sprouts may be strategically used as a climate-smart feed resource for ruminants. However, many stuffs in this manuscript should be improved

The overall English needs improving. The authors should at least find a native English speaker to modify language of the whole manuscript.

Abstract

1. L25, Significant results need to be added with p-values

2. L30, the word of” microbiome diversity” was used inappropriately, which should be modified.

Introduction

1. L69-72, The cost of planting sprouts should be covered and described

Materials and Methods

1. L151, the number of replicates for rumen fluid samples should be added

2. L165, the method of comparing microbial abundance should be described

3. rawdata of microbiota should be deposited into NCBI SRA

Results

1 Figure1, the comparison of methane emissions each day needs to be supplemented and marked with p value.

2 Table2, If the volatile acid is a relative concentration, you need to add a percentage unit to avoid misunderstanding.

3 L275, the microbial similarity comparison between groups should be performed to prove the significant difference in microbial structure with ANOSIM analysis or other methods

Discussion

1. The correlation between signature microbiota and volatile acid, methane emissions and growth performance should be analyzed and discussed in further.

Author Response

Dear Reviewer

Authors would like to thank you for your constructive comments on our manuscript. The comments were address in order to improve the quality of manuscript. hence the manuscript has been updated in consideration of the reviewer's comments.

Thus some parts of the manuscript has been changed particularly in section such as 2.5 was added, 2.6, 2.7, 3.3  were improved as per suggestions. all the changes in the manuscript are highlighted in yellow.

Reviewer 2 Report

The study evaluated the effects of barley sprout on ruminal fermentation characteristics, enteric methane emission and microbiome profiles of meat-master lambs. It has great significance to the efficient utilization of barley sprout forage resources in arid and semi-arid regions. However, there are few aspects of the manuscript that need clarification. 

Overall, the English usage must be reviewed.

 Keywords: Keywords should be consistent with the content of the article. for example, there is no proteolysis-related data in the article. Therefore proteolysis cannot be a keyword for this manuscript. 

Introduction

L62:Adjust the position of (Hordeum vulgare L.) 

L79-81: Pay attention to the expression of this sentence, there is no logical connection with the previous text.

 Material and methods:

L115-117: The chemical composition of the barley sprout not appear in the manuscript. 

L129: Why provide 300g concentrate and what is the basis? 

L132-133: Whether there is any remaining concentrate, if so, it needs to be recorded. 

L132: Provide primer sequences. The polymerase chain reaction (PCR) program should mentioned. 

L165: Provide the production method of Figure 3. 

L171-191: This part belongs to 2.5. No need to write so detailed. 

Results:

L193: Need to provide the nutrients of concentrate, hay and barley sprout. 

L196: The description of the p-value should be as clear as possible. For example, replace p<0.05 with P=0.018. The full text should be revised accordingly. 

L235: A brief description of the sequencing results is required, e.g. how many original sequences were obtained and how many operational taxonomic units (OTU) were these sequences assigned to?

Dilution curves should be provided to demonstrate that the sequencing depth is sufficient and the data are sufficiently reliable.

Are there differences in rumen microbial alpha diversity between treatments?

The relative abundance of rumen microorganisms can be correlated with NH3-NVFA and methane production. 

Table 1: The concentrate, hay and barley sprout intakes should be as detailed as possible 

Table 1: Are the initial body weights of sheep under different treatments the same? 

Discussion:

The discussion must be improved. There was no in-depth analysis of the function of rumen microbes. 

The effect of nutrient intake under each treatment on the relative abundance of rumen microorganisms should be analyzed. 

The relationship between rumen microbes and rumen fermentation parameters and methane production was not analyzed. 

Conclusion:

Conclusion needs to be condensed.

Some content can be put in the discussion section.

Author Response

Dear Reviewer

Thank you very much for your time and constructive comments you made on our manuscript.  The comment were very helpful in improving the manuscript and even for the future study similar tot this one that we may conduct in future.

In improving of the manuscript as per comments from different reviewers the structure (that refers to change of graphs and tables format)  and the numbering of the manuscript changes from the original one.

However to your specific comment we have included and attachment file that respond direct to your comments.  Please see the attachment.

Reviewer 3 Report

Dear authors,

After the review process, I have several comments:

you should include the reference in all Materials and Methods sections; you could include PCoA analysis as a supplementary file;

you should present how was determined the organic acids in section 2;

the comments from section 4 should be correlated with the future valorization of the paper results.

After these comments will be included in the paper, it could be considered for publication.

Best regards!

Author Response

Dear Reviewer

Thank you very much for your time and constructive comments you made on our manuscript.  The comments were very helpful in improving the manuscript and even for the future study similar to this one that we may conduct.

In improving of the manuscript as per comments from different reviewers the structure (that refers to change of graphs and tables format)  and the numbering of the manuscript changes from the original one.

However to your specific comment we have included and attachment file that respond direct to your comments.  Please see the attachment.

Round 2

Reviewer 1 Report

please describe how many rumen samples were collected in M&M.

Reviewer 3 Report

No comments.